Adaptive divergence in resistance to herbivores in Datura stramonium

Castillo Guillermo 1
Valverde Pedro L. 2
Cruz Laura L. 1
Hernández-Cumplido Johnattan 3
Andraca-Gómez Guadalupe 1
Fornoni Juan 1
Sandoval-Castellanos Edson 1
Olmedo-Vicente Erika 1
Flores-Ortiz César M. 4
Núñez-Farfán Juan 1 farfan@unam.mx
1 Department of Evolutionary Ecology, Institute for Ecology UNAM , Mexico Distrito Federal , Mexico
2 Departamento de Biología, Universidad Autónoma Metropolitana-Iztapalapa , Mexico Distrito Federal , Mexico
3 Laboratory of Evolutionary Entomology, Institute of Biology, University of Neuchâtel , Neuchâtel , Switzerland
4 Facultad de Estudios Superiores Iztacala, UNAM , Estado de México , Mexico
Singer Michael
Electronic publication date: 2015 Nov 26
Publication date: 2015
Volume: 3
Electronic Location ID: e1411
Received 2015 May 6; Accepted 2015 Oct 28
Copyright: © 2015 Castillo et al.
Copyright year: 2015
Copyright holder: Castillo et al.
License: This is an open access article distributed under the terms of the Creative Commons Attribution License, which permits unrestricted use, distribution, reproduction and adaptation in any medium and for any purpose provided that it is properly attributed. For attribution, the original author(s), title, publication source (PeerJ) and either DOI or URL of the article must be cited.
License URL: https://creativecommons.org/licenses/by/4.0/

Keywords: Adaptive divergence, Tropane alkaloids, Leaf trichomes, Plant defense, PST–FST comparison, Datura stramonium, Divergent natural selection, Genetic drift and restricted gene flow, Plant resistance

Funding: CONACyT grant 255600 Evolución adaptativa de la defensa en Datura: Resistencia y Tolerancia a los herbívoros PAPIIT-UNAM IN-212214 This study was funded by the CONACyT grant 255600. “Evolución adaptativa de la defensa en Datura: Resistencia y Tolerancia a los herbívoros,” and PAPIIT-UNAM (IN-212214). GC acknowledges the National Council of Science and Technology (CONACyT) for a scholarship and financial support. The funders had no role in study design, data collection and analysis, decision to publish, or preparation of the manuscript.

==============================
Defensive traits exhibited by plants vary widely across populations. Heritable phenotypic differentiation is likely to be produced by genetic drift and spatially restricted gene flow between populations. However, spatially variable selection exerted by herbivores may also give rise to differences among populations. To explore to what extent these factors promote the among-population differentiation of plant resistance of 13 populations of Datura stramonium, we compared the degree of phenotypic differentiation (PST) of leaf resistance traits (trichome density, atropine and scopolamine concentration) against neutral genetic differentiation (FST) at microsatellite loci. Results showed that phenotypic differentiation in defensive traits among-population is not consistent with divergence promoted by genetic drift and restricted gene flow alone. Phenotypic differentiation in scopolamine concentration was significantly higher than FST across the range of trait heritability values. In contrast, genetic differentiation in trichome density was different from FST only when heritability was very low. On the other hand, differentiation in atropine concentration differed from the neutral expectation when heritability was less than or equal to 0.3. In addition, we did not find a significant correlation between pair-wise neutral genetic distances and distances of phenotypic resistance traits. Our findings reinforce previous evidence that divergent natural selection exerted by herbivores has promoted the among-population phenotypic differentiation of defensive traits in D. stramonium.

Introduction

Most species consist of a series of populations that are often phenotypically differentiated (Rice & Jain, 1985; Thompson, 2005). Heritable phenotypic differentiation in multiple traits can be effectively produced by processes like genetic drift, mutation, founder effects or population isolation (Gomulkiewicz et al., 2007). However, phenotypic differentiation in traits that contribute to individuals’ fitness may also have a spatial structure caused by varying selective regimes exerted by biotic and/or abiotic factors (Holsinger & Weir, 2009). Furthermore, stabilizing selection may promote phenotypic similarity among populations (Merilä & Crnokrak, 2001). Elucidating to what extent these processes promote character differentiation among populations is central if we are to fully understand the prevalence of among-population variation in the wild (Lynch, 1990; Althoff & Thompson, 1999; Criscione, Blouin & Sunnucks, 2006; Kelly, 2006; Gomulkiewicz et al., 2007). Here we aimed to determine if among-population variation in traits that confer resistance to herbivores in the annual plant Datura stramonium is consistent with a scenario of varying selection or genetic drift and restricted gene flow.

To infer whether natural selection explains the observed differentiation among populations in putatively adaptive quantitative characters (QST), it is necessary to contrast this hypothesis against a null model of differentiation at adaptively neutral loci (FST; Spitze, 1993; Martin, Chapuis & Goudet, 2008; Whitlock, 2008). The detection of a significant difference between the estimators of differentiation, QST and FST, may imply adaptive differentiation among populations. The comparison of the differentiation indices has three possible outcomes each with a unique interpretation (see Table 1 in Merilä & Crnokrak, 2001). When QST and FST are statistically equal, this implies that the degree of differentiation in quantitative traits could be produced by drift alone. This does not necessarily imply that genetic drift produced the observed phenotypic differentiation but that the roles of selection and drift are indiscernible. When QST < FST, it means that natural selection might be favoring the same phenotype across populations. Finally, when QST significantly exceeds FST, it means that directional selection is favoring different phenotypes in different populations. When QST and FST are equal, it is expected that both indices, estimated among pairs of populations of the same species, will be positively correlated, implying isolation by distance, restricted gene flow and genetic drift (although a partial role of selection could be involved also), or high recombination between molecular neutral marker loci and quantitative trait loci (Merilä & Crnokrak, 2001). In contrast, no correlation between both indices of differentiation among local populations may implicate a role of natural selection (see ‘Discussion’).

In order to explore the signals of non-neutral evolution in quantitative traits it is necessary to estimate QST and FST. FST is commonly estimated by analyzing variance in allele frequency (Wright, 1951) at molecular markers, like microsatellite loci. On the other hand, to estimate QST it is necessary to know the amount of additive genetic variance of quantitative traits in many local populations (Spitze, 1993). However, accomplishing the latter objective is not feasible for a large number of populations because it requires estimating the breeding values of genotypes (families) for a suite of phenotypic characters in each local population. PST (degree of phenotypic differentiation index) is an analogous index to QST (Leinonen et al., 2006; Leinonen et al., 2013), useful for exploring if phenotypic differentiation among populations exceeds genetic differentiation in neutral markers (Merilä & Crnokrak, 2001). The use of PST instead of QST is justified when estimates of additive genetic variance are not available (Leinonen et al., 2006; Leinonen et al., 2013; Lehtonen et al., 2009). Estimation of additive genetic variation in traits makes it necessary to obtain the phenotypic covariance between relatives (families) with an experimental common garden and/or the use of reciprocal transplant experiments to rule out the environmental effects on phenotypes. Hence, PST can be used as a surrogate of QST.

Resistance traits exhibited by plants (i.e., traits that prevent/reduce damage by natural enemies) vary widely across populations (Núñez-Farfán, Fornoni & Valverde, 2007; Züst et al., 2012). Selection exerted by herbivores is a major force driving the evolution of plants’ resistance traits (Rausher, 2001; Anderson & Mitchell-Olds, 2011; Züst et al., 2012). Thus, among-population differentiation in resistance traits is likely to be produced by spatial variation in the local selective regimes exerted by herbivores. Such spatially variable selection can be generated by among-population variation in the abundance, species composition, feeding styles, and degree of dietary specialization of herbivores to their host plants (Falconer & Mackay, 1996; Charlesworth, Nordborg & Charlesworth, 1997; Parchman & Benkman, 2002; Arany et al., 2008; Hare, 2012). Datura stramonium (Solanaceae) provides an optimal system for studying among-population differentiation in resistance traits. Because of its wide distribution (Mexico, Canada, United States, and Europe), D. stramonium is exposed to different environmental conditions and to a wide diversity of herbivore species (Weaver & Warwick, 1984; Valverde, Fornoni & Núñez-Farfán, 2001; Cuevas-Arias, Vargas & Rodríguez, 2008). Resistance against herbivores in D. stramonium includes leaf trichomes (Valverde, Fornoni & Núñez-Farfán, 2001; Kariñho-Betancourt & Núñez-Farfán, 2015) and tropane alkaloids (Shonle & Bergelson, 2000), of which atropine, hyosciamine and scopolamine are the most abundant (Parr et al., 1990; Kariñho-Betancourt et al., 2015). These secondary metabolites affect the activity of the neurotransmitter acetylcholine (Roddick, 1991) with negative effects on insects and vertebrate herbivores (Hsiao & Fraenkel, 1968; Krug & Proksch, 1993; Wink, 1993; Shonle, 1999; Mithöfer & Boland, 2012). Recent studies have found ample geographic variation in leaf trichome density and atropine and scopolamine concentration in central Mexico (Castillo et al., 2013; Castillo et al., 2014). However, it is unclear if selection by herbivores or neutral processes, among other factors, can account for the observed among-population differentiation in these resistance traits.

Here, we assessed to what extent population differentiation in resistance leaf traits (trichome density, atropine and scopolamine concentrations) of D. stramonium is accounted by neutral processes (genetic drift and restricted gene flow) or divergent natural selection. To do so, we compared the degree of phenotypic differentiation of resistance traits by means of PST estimated for the whole range of values of heritability, with the neutral expectation set by allelic divergence at microsatellite loci (FST). We expect that PST of each resistance character would be significantly higher than the index of population differentiation in neutral molecular markers (FST), since previous studies have detected contrasting selection exerted by herbivores on the three characters.

Methods

Study system

Datura stramonium L. (Solanaceae) is an annual herb commonly found in roadsides, cultivated areas and disturbed environments in Mexico, the United States, Canada, and Europe (Valverde, Fornoni & Núñez-Farfán, 2001; Weaver, Dirks & Warwick, 1985; Van Kleunen, Markus & Steven, 2007). In Mexico, leaves of D. stramonium are consumed by a dietary specialist herbivore, the chrysomelid Lema trilineata (Nuñez-Farfan & Dirzo, 1994), the dietary oligophagous Epitrix parvula (Chrysomelidae), which also feeds from other members of the Solanaceae family (Glass, 1940), and by the dietary generalist grasshopper Sphenarium purpurascens (Nuñez-Farfan & Dirzo, 1994). Datura stramonium features leaf trichomes and tropane alkaloids (atropine and scopolamine) as resistance traits against herbivory. These traits have shown heritable basis (Shonle & Bergelson, 2000; Valverde, Fornoni & Núñez-Farfán, 2001; Kariñho-Betancourt & Núñez-Farfán, 2015), and are under selection by dietary specialist and generalist herbivores (Castillo et al., 2014).

Fieldwork

During August–September 2007 we sampled 13 natural populations of D. stramonium in central Mexico (Fig. 1). Selected populations inhabit a wide range of habitat types. The geographic location and habitat characteristics are shown in Table 1. From each population we sampled 30 randomly chosen individual plants.

Figure 1 Sampled populations of Datura stramonium in central Mexico (see Table 1).

Table 1 Vegetation type, latitude, longitude, altitude and population means of leaf trichome density, and atropine and scopolamine concentrations of 13 populations of Datura stramonium in central Mexico.

	Vegetation type	Latitude	Longitude	Altitude (m a.s.l.)	Trichome density (2.5 × mm2)	Atropine (mg/g)	Scopolamine (mg/g)	
1. Acatzingo	DS	−97.78	19.32	2,160	8.99	0.295	0.159	
2. Atlixco	DS	−98.42	18.98	1,840	9.04	0.691	0.577	
3. Esperanza	DS	−97.37	18.85	2,278	9.57	0.535	0.542	
4. Patria Nueva	DS	−98.96	20.38	2,040	12.62	0.317	0.367	
5. Taxco	TDF	−99.66	18.5	1,582	9.02	0.957	0.266	
6. Teotihuacán	DS	−98.86	19.68	2,294	8.73	0.437	0.353	
7. Ticumán	TDF	−99.2	18.86	1,210	6.6	0.938	1.889	
8. Tlaxiaca	DS	−98.86	20.08	2,340	9.36	0.288	0.458	
9. Tula	DS	−99.35	20.05	2,020	6.06	3.129	2.804	
10. Tzin Tzun Tzan	POF	−101.58	19.63	2,050	4.29	0.994	2.995	
11. Valsequillo	DS	−98.11	18.91	2,209	6.09	1.767	0.044	
12. Xalmimilulco	POF	−98.38	19.2	1,200	4.66	2.688	2.513	
13. Zirahuén	POF	−101.91	19.43	2,174	4.91	0.618	1.968	
Notes.

DS desert shrub

POF Pine–Oak forest

TDF tropical deciduous forest

Resistance traits quantification

Following Valverde, Fornoni & Núñez-Farfán (2001), we estimated leaf trichome density as the total number of trichomes in an observation field of 2.5 mm2 located in the central basal region of the adaxial side of the leaf, using a stereoscopic microscope. Then we averaged the trichome density per plant from a random sample of 20 fully expanded leaves. We also quantified the concentration of atropine and scopolamine (two major alkaloids in D. stramonium) from a sample of 20 leaves per plant by means of High Precision Liquid Chromatography (HPLC). Details of the extraction method and HPLC conditions can be found elsewhere (see Castillo et al., 2013).

Data analysis

We estimated the neutral genetic differentiation among populations of D. stramonium using FST values obtained from five nuclear microsatellite markers designed specifically for D. stramonium as reported by Andraca-Gómez (2009). FST values were calculated using FSTAT 2.9.3.1 (Goudet, 2001) employing approximately 30 individuals per population. In addition, we assessed the statistical power of our five microsatellites by means of Wright–Fisher simulations as implemented in the program PowSim (Ryman & Palm, 2006). The program requires a divergence time and effective populations sizes so we tested a number of feasible combinations.

Phenotypic divergence in resistance traits

We used the degree of among-population phenotypic divergence (PST) to explore if restricted gene flow and genetic drift (FST) alone can account for this differentiation or if there is a signal of differentiation promoted by divergent selection on resistance traits (Leinonen et al., 2006; Pujol et al., 2008). We estimated PST as PST=σGB2σGB2+2h2•σGW2,

where σGB2 is the variance among populations, σGW2 is the variance within population, and h2 is the trait heritability (Leinonen et al., 2006). Since this is not feasible for a large number of populations we used an approximation by PST.

In order to obtain PST values for resistance traits, we simulated the whole range of heritabilities (0 ≤ h2 ≤ 1). To estimate PST values we fitted a linear model for each resistance trait, under the assumption that the distribution of resistance traits was normally distributed. The population term was considered as a random effect. To test the hypothesis that PST is higher than FST, a Monte Carlo test was carried out, approaching a sample of 10,000 deviates from both PST and FST by means of their estimated error. PST error was estimated from the likelihood errors of its components (variances among- and within-populations), while FST error was obtained by bootstrapping (Goudet, 2001). The 10,000 random deviates of FST and PST were compared and the p-value was obtained as the proportion of comparisons in which the FST was equal or higher than the PST (null hypothesis).

We further evaluated the pair-wise Pearson’s correlation between FST and PST for all populations. Neutral marker variation can be used as a neutral expectation against which the phenotypic divergence of traits can be compared (Gomulkiewicz et al., 2007). If resistance phenotypic differentiation between populations (PST) is the result of neutral processes rather than selection, differentiation among populations in these traits should correlate positively with differentiation in selectively neutral markers (FST) (Merilä & Crnokrak, 2001; Gomulkiewicz et al., 2007; Lehtonen et al., 2009; Leinonen et al., 2013). We evaluated the pair-wise correlation between the FST and PST for different scenarios of heritability (h2 = 0.1, 0.25, 0.5, 0.75 and 1.0). Statistical analyses were performed using JMP® version 9.0.0 (SAS Institute, Cary, NC, 1989–2007).

Results

Among-population variation in resistance traits

A multivariate analysis of variance (MANOVA) detected significant multivariate differences in the studied resistance traits of 13 populations of D. stramonium (Wilks’ λ = 0.091, F36,331.64 = 11.51, P < 0.0001). After the subsequent univariate ANOVAs were applied, we found significant differences in trichome density (F12,126 = 5.10, P < 0.0001), atropine (F12,126 = 7.85, P < 0.0001), and scopolamine concentration (F12,126 = 23.33, P < 0.0001). Mean leaf trichome density and mean atropine and scopolamine concentration per population are shown in Fig. 2 and Table 1.

Figure 2 Among-populations variation in leaf trichome density (A), and atropine and scopolamine concentration (B) in 13 populations of Datura stramonium in central Mexico.

Bars represent average value +1 SE.

Genetic differentiation between populations of D. stramonium

Genetic differentiation as estimated by differences in allele frequency at microsatellite loci was moderate. FST was 0.228 (S.E. = 0.039), which is well above the minimum detectable value (FST = 0.01) that our sample and markers allowed with a statistical power of 0.94.

Phenotypic divergence in resistance traits

Comparison of phenotypic (PST) and neutral genetic marker divergence (FST) showed that PST values for scopolamine concentration were significantly higher than the FST in all values of h2 (Fig. 3). However, PST for atropine concentration was significantly higher than FST when 0 ≥ h2 ≤ 0.3 (Fig. 2), whereas PST of leaf trichome density significantly exceeded FST only when h2 ≤ 0.1 (Fig. 2).

Figure 3 PST values of putative defensive traits of Datura stramonium as a function of their genetic variance (h2) among populations.

Confidence intervals of 50% and 95% are indicated by bars and lines, respectively. * Represents overall PST values that differ significantly from FST (the black bar at the right end) after a Monte Carlo test (10,000 deviates from both PST and FST; see ‘Methods’).

Pair-wise correlation between FST and PST

We found no significant correlations between pair-wise FST and PST values among populations for any of the three resistance characters (Table 2). Most correlation values were small (i.e.,−0.135 ≤ r ≤ 0.034).

Table 2 Correlation (r) between pair-wise PST of three resistance traits and pair-wise FST for all populations of Datura stramonium, under different scenarios of heritability (h2 = 0.1, 0.25, 0.5, 0.75 and 1.0).

Resistance trait			r			
	h2 = 0.1	h2 = 0.25	h2 = 0.5	h2 = 0.75	h2 = 1.0	
Atropine	−0.0644	−0.0671	−0.0655	−0.0642	−0.0637	
Scopolamine	0.0264	0.0344	0.0348	0.0316	0.0278	
Trichome density	−0.135	−0.1218	−0.1053	−0.0939	−0.0855	

Discussion

Results showed that phenotypic differentiation in resistance traits among population of D. stramonium is not consistent with divergence promoted by genetic drift and restricted gene flow alone (Pujol et al., 2008; Lehtonen et al., 2009). Phenotypic differentiation in scopolamine concentration was significantly higher than FST across the range of h2. In contrast, genetic differentiation in trichome density was different from FST only when heritability was very low, and most phenotypic variation could be related to major environmental factors, like annual mean precipitation and temperature. Likewise, differentiation in atropine concentration seems to differ from the neutral expectation only at low values of h2. Furthermore, we did not find a correlation between pair-wise neutral genetic distances and phenotypic distances of any of the three resistance traits. Taken together, results suggest that natural selection could be involved in phenotypic divergence on resistance traits among populations of D. stramonium.

Results indicate that populations of D. stramonium are differentiated in both phenotypic and neutral molecular markers. We found a moderate amount of differentiation among populations at microsatellite loci (FST = 0.228). Using this FST value, the indirect estimate of gene flow (Nm) is 0.846, suggesting restricted gene flow among populations of D. stramonium, and not sufficient to prevent differentiation by genetic drift (Hedrick, 2000). This contrasts with differentiation at neutral loci reported for other organisms where FST is generally lower than 0.228 (but see Merilä & Crnokrak, 2001). PST index values statistically not different from this value of FST imply that quantitative phenotypic characters follow a pattern of drift-induced divergence (Leinonen et al., 2006). Here, we found that the PST index of scopolamine was significantly higher than FST for all values of heritability considered (cf. Fig. 3). This result strongly suggests that phenotypic differentiation among populations in scopolamine concentration is congruent with a scenario of divergent selection exerted by herbivores among populations. However, PST of atropine and leaf trichome density was higher than FST only when heritability was ≤0.3 and ≤0.1, respectively. This implies that the proportion of genetic variance among populations from total genetic variance is high for these characters (Lehtonen et al., 2009; Leinonen et al., 2013). When genetic variance within populations is low, as implied by low values of heritability, there is a high opportunity to detect a significant PST given that the among-population genetic variance component has a relevant weight in the total phenotypic variance. Inversely, when heritability is high, the within-population genetic component accounts for a high fraction of total genetic variance rendering PST very small. These considerations may explain why PST of trichome density and atropine are different from FST only at very low heritability.

Although PST is used as an analog of QST (genetic differentiation in quantitative characters) when it is not possible to obtain the amount of additive genetic variation (variance among families, within populations) (Merilä & Crnokrak, 2001), conclusions derived from these estimations must be interpreted with caution since this index can be biased by all environmental variation due to abiotic conditions among localities as well as environmental deviations within populations, and non-additive genetic variation (v.gr., epistatic interactions, dominance, linkage disequilibrium), among others (Pujol et al., 2008). Thus is relevant to ask whether PST index obtained for the resistance traits in D. stramonium posses genetic variance. Datura stramonium displays a great variation among populations in trichome density and tropane alkaloids’ concentration in central Mexico (Castillo et al., 2013). Phenotypic variation in alkaloid concentration, like other quantitative traits, is governed by environmental physical factors and genetic variation (Castillo et al., 2013). Previous studies in this species have detected narrow-sense h2 of general resistance to herbivores of 0.49 and 0.41 in two natural populations of D. stramonium (Fornoni, Valverde & Núñez-Farfán, 2003; note that general resistance may include physical and chemical defenses). In addition, broad-sense h2 of general resistance and trichome density has been estimated in 0.25 and 0.64, respectively (Kariñho-Betancourt & Núñez-Farfán, 2015). Also, genetic variance in trichome density among-populations (Valverde, Fornoni & Núñez-Farfán, 2001) and general resistance (Valverde, Fornoni & Núñez-Farfán, 2003; Carmona & Fornoni, 2013) has been detected in D. stramonium. Finally, genetic variance in alkaloid concentration (hyosciamine and scopolamine, and their ratio) has been detected previously by Shonle & Bergelson (2000). Thus, there is ample evidence of genetic basis of phenotypic variation in resistance of D. stramonium to support our estimation of PST values.

Because a PST index higher than FST means that divergent selection might be involved in population differentiation of resistance traits, at least for scopolamine, it is relevant to ask to what extent natural selection by herbivores is responsible for population differentiation in this character. In D. stramonium, several lines of evidence strongly suggest that differentiation in resistance is accounted by for herbivores. Differential and contrasting selection gradients on resistance to herbivores were detected between two populations of this species in a reciprocal transplant experiment (Fornoni, Valverde & Núñez-Farfán, 2004). Likewise, Shonle & Bergelson (2000) detected stabilizing selection on hyosciamine and directional selection to reduce scopolamine concentration in D. stramonium. In a recent study of eight populations of D. stramonium, Castillo et al. (2014) found that atropine is selected against by the dietary specialist herbivores Epitrix parvula (in one population) and Lema daturaphila (in two populations). In contrast, scopolamine was positively selected in one population where the specialist Lema daturaphila was the main herbivore, whereas trichome density was positively selected in two populations (one with L. daturaphila and one with the generalist grasshopper Sphenarium puprurascens), and negatively selected in one population with the E. parvula (Castillo et al., 2014). Thus, although genetic drift and restricted gene flow could produce phenotypic variation in plant resistance among populations, the available evidence of spatially variable selection on resistance traits in D. stramonium and data presented here suggests that population differentiation can be potentially adaptive.

Furthermore, we did not detect any significant correlation between the pair-wise PST and FST among population across the whole range of heritability, suggesting that differentiation at quantitative traits and neutral molecular loci is decoupled. Theoretically, if the pace of differentiation is dictated by genetic drift only, it is expected that differentiation indices will be perfectly and positively correlated (r = 1, PST = FST; Fig. 4). If the correlation is positive but lower than 1, then genetic drift has a role but does not explain all differentiation in quantitative traits. In the region above the diagonal in Fig. 4, where PST > FST, any positive pair-wise correlation across populations, depicts a scenario where differentiation in quantitative traits exceeds the neutral expectation and suggests divergent selection (Fig. 4). On the other hand, in the region below the diagonal, where PST < FST, any positive pair-wise correlation across populations, portrays a scenario where differentiation at neutral molecular loci surpasses that of quantitative characters suggesting a strong effect of genetic drift; however at moderate values of FST stabilizing selection might be favoring the same phenotype across populations (Fig. 4). When PST > FST and are uncorrelated (dotted line in Fig. 4) it shows another interesting scenario, as found here. This implies that genetic drift and restricted gene flow alone cannot explain (Pujol et al., 2008) the pattern of differentiation among populations in resistance traits in D. stramonium. Under this scenario there is opportunity for divergence driven by selection in resistance traits. Our results suggest that the higher PST than FST for scopolamine, together with spatial variation in resistance traits and the existence of a selection mosaic detected previously by Castillo et al. (2014) are consistent with outcomes predicted by the geographic mosaic of coevolution (Thompson, 2005).

Figure 4 Theoretical relationship between pair-wise PST and pair-wise FST across populations of a species.

The solid diagonal line indicates a perfect and positive correlation between both indices (r = 1, PST = FST). Above the diagonal, blue points are pairs of populations where PST > FST. Below the diagonal, orange points are pairs of populations where PST < FST. Dotted line indicates one possible scenario where both indices are uncorrelated. At moderate values of FST stabilizing selection might be promoting low phenotypic differentiation between a given pair of populations (big orange point).

Supplemental Information

Supplemental Information 1 Defensive characters in Datura stramonium

Data of trichome density, atropine and scopolamine concentration in plants of Datura stramonium from 13 populations in Central Mexico.

Click here for additional data file.

We thank very much to Professor Michael Singer for his valuable revision to our manuscript. We thank to Blanca Hernández, Martha Macías Rubalcava, María Teresa Caudillo, Luis Barbo and Martha Urzúa Meza for helping us during HPLC quantification, and to Rosalinda Tapia-López and the members of Laboratorio de Genética Ecológica y Evolución for their logistical support and field assistance. Thanks are also extended to the Laboratorio de Alelopatía of Instituto de Ecología, UNAM for providing the facilities for laboratory work. This paper constitutes a partial fulfillment of the Graduate Program in Biological Sciences of the National Autonomous University of Mexico (UNAM).

Additional Information and Declarations

Competing Interests

Author Contributions

Data Availability

The authors declare there are no competing interests.

Guillermo Castillo conceived and designed the experiments, performed the experiments, analyzed the data, wrote the paper, prepared figures and/or tables, reviewed drafts of the paper.

Pedro L. Valverde analyzed the data, wrote the paper, reviewed drafts of the paper.

Laura L. Cruz, Guadalupe Andraca-Gómez and Juan Fornoni performed the experiments.

Johnattan Hernández-Cumplido performed the experiments, reviewed drafts of the paper.

Edson Sandoval-Castellanos analyzed the data, wrote the paper, prepared figures and/or tables, reviewed drafts of the paper.

Erika Olmedo-Vicente analyzed the data.

César M. Flores-Ortiz performed the experiments, contributed reagents/materials/analysis tools.

Juan Núñez-Farfán conceived and designed the experiments, analyzed the data, contributed reagents/materials/analysis tools, wrote the paper, reviewed drafts of the paper.

The following information was supplied regarding data availability:

The research in this article did not generate any raw data.

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
