# Peer review of "Adaptive divergence in resistance to herbivores in Datura stramonium"

_PeerJ, doi:10.7717/peerj.1411_

## Round 0.1 · original submission · Major Revisions

The reviewers and I recognize the value of your study, and we agree on several substantial changes needed to make this manuscript publishable.

First, you will need to provide justification in the Methods for the use of 0.5 and 1 as heritability values for calculating Pst.

Second, you will need to provide justification for the adequacy of only 5 genetic markers for calculating Fst.

Third, you will need to discuss the limitations of or show further evidence for your argument that the phenotypic variation in defensive traits is under selection from herbivores. The reviewers rightly raise the possibility that other agents of selection might be responsible for phenotypic variation in these traits. Having read your 2014 PLOS ONE paper, I think you could make a stronger connection between the analysis in that paper and the analysis in this paper. I leave it up to you to decide how to do this: adding further quantitative analysis versus discussing the limitations of the current analysis. I agree with Reviewer 1 that a quantitative test is needed to make a strong case for herbivores acting as agents of selection on these traits.

Lastly, you will need to address each of the other comments from the reviewers in your rebuttal letter. All of the reviews are thorough and constructive, with great suggestions for improving the manuscript.

Reviewer 1 ·

Basic reporting

Carillo et al. present a comparative study of phenotypic vs. neutral genetic differentiation among 13 natural populations of the plant Datura stramonium in central Mexico. They measure physical and phytochemical traits associated with defense on a sizeable number of field-collected plants in each population, and estimate Fst values from 5 microsatellite marker in all plants. If a trait shows higher phenotypic differentiation than expected based on Fst, they interpret this as the result of adaptive local adaptation, in particular in response to herbivores.

As a minor comments, the information in Table 2 seems redundant with the information contained in Figure 2, and puts a lot of emphasis on mostly meaningless p-values. Four decimal places also imply higher precision than can be expected from simulation (Monte Carlo) data.

Experimental design

The authors sampled a large number of populations with good replication, thus the study design seems appropriate. However, the authors only use 5 microsatellite markers to generate their Fst values, which seems low in comparison to similar studies. I would expect a justification of why 5 markers is sufficient to characterize genetic divergence in Datura.

Unfortunately I’m not convinced that the analysis of phenotypic divergence performed by the authors is particularly informative. The authors use Pst to describe among-population phenotypic divergence and state that “the use of Pst instead of Qst is justified when estimates of the latter are not available” (L107). This seems like a weak argument, and the authors do not further discuss potential problems of Pst. I assume estimation of Qst would require measuring phenotypes in a common environment or working with known genotypes/families, both of which would of course require more effort than may be feasible for a relatively small study such as this one. Still, the authors should then provide clear support for why Pst is a valid measure.

As it stands, Pst is a ratio of the among-population variance vs. within-population variance, which is weighted by trait heritability. The lower the heritability, the larger the within-population variance can be relative to among-population variance to maintain a high value of population differentiation. Using an appropriate value for heritability is thus key for getting meaningful results, with higher values of heritability providing more conservative estimates of Pst.
However, instead of estimating heritabilities for their plant traits or inferring it from other studies, the authors choose seemingly arbitrary values of 1 and 0.5 and state that this reflects “[…] two plausible scenarios of heritability” (L179). It is unclear to me how this claim is supported, as for any plant system a heritability value of 1 seems highly unrealistic. I assume a value of 1 could be seen as the most conservative, but intentionally unrealistic scenario, but in that case this should be specified accordingly. The fact that the authors find a significant effect on Scopolamine even at the most conservative value of 1 indicates that this trait may truly be differentiated among populations, but for the other two traits interpretation is less clear as there is no biological justification for a specific value of heritability.

Validity of the findings

The main result of the study is that one chemical trait shows signs of adaptive divergence among populations, while the other two traits may potentially show similar patterns depending on trait heritability. In the third paragraph of the discussion, the authors set up these result by citing several studies that demonstrate how herbivore selection drives plant defensive traits. This structure insinuates that the present study shows the same mechanism at work, yet the authors don’t actually test this. The sampled populations without a doubt differed in any number of factors, both abiotic and biotic. None of these factors were accounted for, thus I would expect some discussion of other potential driving forces. Plant secondary chemistry has many functions in plant and isn’t exclusively defensive, and this is even more the case for trichomes. Any one of these functions might be the cause for the observed adaptive, non-neutral divergence. Have the authors considered including climatic variables as potential drivers of the patterns in their data?
The authors mention that they found differential selection by generalist and specialist herbivores on secondary chemistry of Datura. However, from reading the manuscript it is unclear to me whether specialist and generalist herbivores would actually show sufficient variation in their relative abundance to be responsible for such population differentiation.

Reviewer 2 ·

Basic reporting

Overall I think the paper is written well but I have a few suggestions that I think will improve the paper:

1) Lines 47-49: I think it would be helpful to provide an example from the literature where variation in abiotic factors influence evolutionary and local adaptation in plant defenses.

2) Line 57: I recommend that atropine and scropolamine are defined as being defensive alkaloids here or perhaps just described as two chemical defenses.

3) Line 75: typically the convention is to spell out the genus when it is at the start of a sentence.

4) Line 132: remove 'leaf damage' from paragraph title

5) Line 139: The text refers to Table S2 but my materials to review did not have this table and there does not seem to be a Table S1 either.

6) Line 150: remove first 'overall' from sentence.

7) Line 184: the line "traces of genetic among-population differentiation" is vague and suspect. Please be more specific.

Experimental design

The experimental design is solid and robust but I do still have some concerns with the use of traits measured in the field and not a common garden. I still do not understand if the approach used to deal with this issue was appropriate.

8) Line 113-114: I was left wondering why the authors chose these two particular scenarios (i.e. genetic effect accounting for half and all the variation). Is it based on known heritabilities for these traits, or average heritabilities, or just two points to represents a wide amount of variation? Some justification for the approach would give it more credibility.

9) Line 90: can the authors provide any data on how plastic these traits are to give an expectation of how much they are influenced by environmental factors?

10) On a different and more minor note, I recommend the authors include what R packages and functions where used to conduct each of their analyses. I realize at this point this is a matter of personal preference but I think the readers typically appreciate some greater detail in what specific functions are used to conduct analyses.

Validity of the findings

For the most part I think the results are interpreted appropriately. A study of this sort runs the risk of falsely inferring process from pattern. In most cases the language in this paper does not make this mistake. However I think there could be some more discussion of alternate hypothesis rather than just herbivory alone driving patterns:

11) Line 173-176: I think the discussion would be strengthened by a mention of some alternate hypotheses that could lead to spatial variation in defenses, for example trade-offs between abiotic and biotic stressors. Also, an additional analysis that could easily be included using data presented in the paper is to test if altitude, habitat type, latitude, or longitude predict any variation in defenses among populations.

12) I think the discussion should include an interpretation for atropine Pst being higher than Fst when H2=0.5 but not H2 = 1. The meaning and implications of this are not explained.

Additional comments

13) I disagree with sentence at Line 164-166: "few studies have explored whether natural enemies constitute a predominant force promoting geographic variation on plant defense." To name a few, studies like Zangerl and Berenbaum 2005 PNAS, Stenberg et al 2006 Oecologia, Lennartsson et al 1997 Am Nat, and work by Anurag Agrawal on Milkweeds and a bunch of work by Jon Agren, also a huge number of studies that date back to min 1900's on impacts of grazing mammals on plant local adaptation, all are looking at the impacts of herbivores on spatial variation in defenses. I recommend a revised statement that focuses on a more specific area of study and to more accurately reference previous work that focuses on similar questions.

Reviewer 3 ·

Basic reporting

This paper examines whether population differentiation in physical and chemical defensive traits is due to natural selection or neutral processes such as genetic drift. Previous work on the Datura stramonium system has indicated that there is spatially variable selection on defense traits by herbivores; this paper extends this research by testing for signatures of selection on 13 different populations of D. stramonium.

One issue that needs to be addressed throughout the paper is whether herbivores are actually causing selection on the defense traits being studied. For example, trichomes are known to increase water-use efficiency by providing a barrier between the leaf surface and the environment. Likewise, chemical defenses may have other, non-defensive functions or may be correlated with other traits (and thus be under correlated selection). Despite this, in areas of the manuscript such as line 58, the authors make it seem like natural selection by herbivores vs. neutral processes are the only two options. In the introduction, the authors need to better explore (or convince the reader otherwise) that selection on plant compounds or structures might be due to factors other than herbivores. Similarly, in lines 175-176, the authors need to provide alternative hypotheses for observed patterns of selection on defensive compounds, if such hypotheses exist. Since the authors are unable to differentiate between herbivores and other agents of selection using their particular methods, they must account for this.

A second issue is that on Line 132, the heading includes “leaf damage”, but the authors do not mention measuring this in either the methods or results. If they did measure leaf damage, it seems like it would be important for several reasons: a) it would suggest whether there is spatial variation in herbivory (required for differential selection), b) the chemical and physical defenses might be plastic in response to damage, increasing non-genetic variation, and c) differences in damage might suggest differences in resistance to herbivores, an additional “defense” trait. Therefore, “leaf damage” should be either deleted or discussed in the context of the results and included in the methods.

Experimental design

No Comments.

Validity of the findings

While the results of this work are suggestive that D. stramonium chemical defense traits are evolving in a non-neutral manner, the authors’ conclusions would be stronger with better justification of their statistical and experimental methods. One thing that they need to more strongly justify is their use of 0.5 and 1.0 as heritability values when calculating Pst. Do the authors have heritability estimates from previous experiments or are able to find values in relevant literature? If they are under-estimating heritability (i.e., by using 0.5 as the lower cutoff), this will make it more likely to see a significant difference between Pst and Fst in their analysis (Table 2), particularly for atropine.

Another method that needs to be better justified is the number of microsatellite markers used to calculate Fst. While there isn’t a hard and fast rule that I’m aware of, 5 loci is a small number of markers in comparison with the other studies that use similar methods and that the authors cite (Lehtonen et al. 2009, Leinonen et al. 2006, which use 19 and 18 loci, respectively). As the authors are basing most of their analyses on their Fst values, if these values are estimated poorly, it could have a large impact on their conclusions. Specifically, if the authors are under-estimating the Fst values, it will make it more likely that they find erroneous differences between Pst and Fst for their traits. One thing that would increase confidence in their results is if the authors provided more information on the specific loci they used in their study. How many alleles are there at each microsatellite locus? How well is each allele represented? Using only 5 loci makes it more likely that the pattern is being driven by a lack of variation in 1-2 loci, and may not be representative of the amount of netural differentiation among populations.

Additional comments

Line 39: “plants” needs to be possessive
Line 167: multiple spelling errors (shown, through, and soil types)

---

## Round 0.2 · Minor Revisions

I sent the revised manuscript back to one of the original reviewers because my reading indicated that it still needed some minor revision. This new review reinforces my thoughts about it.

In your minor revision, please follow closely the constructive comments from the reviewer. It will be important for you to clarify the key question (distinguishing between neutral evolution and selection) near the beginning of the Introduction section, and to use the rest of the Introduction section to discuss this issue. How exactly does your approach address this question (rather than other questions, such as sources of biotic selection)?

The idea goes for the Discussion section. Discuss how exactly your evidence addresses the key question. The Discussion section is the best place to expand on the evidence from other studies that argue for herbivores as agents of selection for the traits under study here.

I suggest that you use some of the text from your first rebuttal letter to provide further justification and explanation in parts of the manuscript. There might have been some miscommunication during my previous letter (first review). I apologize if I was not clear. My request to provide further justification and explanation in places was a request for revising the manuscript, not merely providing explanation in the rebuttal letter. Remember that it is ultimately the readers you need to convince, not just these reviewers and me, and an online journal like PeerJ does not impose such severe space limitations as a print journal. Use this to the advantage of your science!

Minor comments:
Remove the comma after "crucial" in line 33.

In the paragraph beginning with line 36, replace "defensive traits" with "resistance traits." Note that in the plant defense literature, resistance and tolerance are two general strategies, so you can use these terms without having to define your meaning of "defensive." Make the same change throughout the manuscript (e.g., lines 71, 74).

In your description of the study system, use "dietary generalist" or "dietary specialist" rather than "generalist" or "specialist." It is important to be clear about what type of specialization you mean.

Remove the comma after "stramonium" in the first sentence of the Discussion.

In line 221, replace "is" with "might be" and in line 222, remove "be".

Reviewer 1 ·

Basic reporting

I found parts of the manuscript by Castillo et al. much improved, particularly the methods and the results section that I have no more issues with. However, the discussion does not actually help that much at all in understanding the potential meaning behind these results, and several points the authors make seem to be claims without indication how they are supported by the data.

A few minor points:
L52: The whole paragraph deals with different biotic natural selection, but then concludes again with the difference between herbivore selection and neutral processes. See point below on what this data can be used for.
L115: add ‘alone’ after ‘(Fst)’
L116: change to ‘selection on defense…’
L124: Move the sentences starting with ‘Estimation of additive…’ back before ‘We estimated Pst…’
L173: remove comma after ‘stramonium’, add ‘alone’ after ‘gene flow’.
L178: Odd to introduce this sentence with ‘on the other hand’ when the previous sentence discussed a similar finding (significance only at low heritabilities).

Experimental design

No more comments

Validity of the findings

First, the manuscript isn’t really clear on what the key question is. The major question that can be answered with the approach taken by the authors seems to be whether patterns in defensive traits of Datura are due to drift, or due to natural selection. However, much of the introduction and the discussion are devoted to distinguishing between different forces of biotic natural selection. I don’t really see how this type of data could answer this, but the authors seem to imply as much (e.g., L213). If it is really true that this can be gleaned from the data this has to be explained much better. If not, all such statements should be removed.

While I like the results from Figure 3, I expect a discussion of what this means, exactly. How is it relevant that Pst differs from Fst mostly at low heritability values? What would the priori expectations have been? Does this allow for any deductions on the types of driving selective forces? The authors mention some actual values of heritability later in the discussion (that are mostly outside of this range), but this is not tied in with these results. The paper is not long, so there really is no reason not to have a more detailed discussion of what this all means.

What would a pairwise correlation between Pst and Fst mean, exactly? The authors mention the absence of such a correlation, but do not explain why this is important or meaningful. Also, the authors maintained just two levels of heritability, 0.5 and 1 for this correlation. How does this tie in with two of the three traits only showing significant differences between Pst and Fst at heritability values that are lower than this?

---

## Round 0.3 · Minor Revisions

I am pleased to see the revisions of this new version of the manuscript. It looks very strong except for some minor grammatical issues. Therefore, I did not need to send it out for review yet again. Instead I reviewed it carefully, so that it will be ready for publication after this very minor revision. Please make the last minor revisions according to my attached edits.

---

## Round 0.4 · accepted · Accept

Great work making the necessary revisions!